# Generalizability of Machine Learning to Categorize Various Mental Illness Using Social Media Activity Patterns

## Chee Siang Ang and Ranjith Venkatachala *

School of Computing, University of Kent, Canterbury CT2 7NB, UK
* Correspondence: rv231@kent.ac.uk

**Abstract:** Mental illness has recently become a global health issue, causing significant suffering in people's lives and having a negative impact on productivity. In this study, we analyzed the generalization capacity of machine learning to classify various mental illnesses across multiple social media platforms (Twitter and Reddit). Language samples were gathered from Reddit and Twitter postings in discussion forums devoted to various forms of mental illness (anxiety, autism, schizophrenia, depression, bipolar disorder, and BPD). Following this process, information from 606,208 posts (Reddit) created by a total of 248,537 people and from 23,102,773 tweets was used for the analysis. We initially trained and tested machine learning models (CNN and Word2vec) using labeled Twitter datasets, and then we utilized the dataset from Reddit to assess the effectiveness of our trained models and vice versa. According to the experimental findings, the suggested method successfully classified mental illness in social media texts even when training datasets did not include keywords or when unrelated datasets were utilized for testing.

**Keywords:** mental health; social media; machine learning; Twitter; Reddit





## 1. Introduction

The biopsychosocial approach recognizes that health and illness are a result of complex interactions between psychological, biological, and social factors. This viewpoint (biopsychosocial approach) considers health to be a condition of mental, physical, and social well-being, rather than simply the absence of illness [1]. The biopsychosocial approach acknowledges that social, environmental, cultural, and socioeconomic factors, as well as biological aspects such as genetics and physiology, and psychological aspects such as emotions, attitudes, and behaviors, all have an impact on physical health. In contrast to the common misconception that mental health is merely the absence of mental illness, the biopsychosocial approach to mental health and illness views mental health as a state of general psychological well-being. Mental illness is a complicated and varied phenomenon that is influenced by a range of psychological, biological, and social factors. It is frequently characterized by changes in thoughts, emotions, behaviors, and social functioning, all of which significantly impair daily functioning and cause severe discomfort [2].

Mental illnesses refer to "a wide range of mental health conditions that affect mood, thinking, behavior, and relationships with others", and pose devastating threats to personal well-being [3]. Mental illness can range from a variety of conditions, including depression and personality disorders such as borderline personality disorder (BPD), bipolar disorder, schizophrenia, anxiety, and drug or alcohol use disorders [4]. In the United Kingdom (UK), mental diseases cost around GBP 105 billion annually. Mental diseases are the biggest cause of sickness absence in the UK, accounting for 70 million sick days lost each year. A total of 44% of applicants for employment and support assistance have a mental condition as their primary diagnosis [5,6]. According to predictions, mental illness could have a global economic impact of more than USD 5 trillion by 2030 [7]. It is estimated that more than 792 million people of all ages worldwide suffer from mental health problems, according to the World Health Organization's (WHO) Fact Sheet published in 2017 [8].

Szasz's view was that mental illnesses are not medical conditions, but difficulties that arise from social, cultural, and psychological influences. Szasz contended that classifying individuals as mentally ill and treating them, as well as using other medical measures, are methods of social regulation that restrict personal independence and self-determination. In lieu of this, Szasz supported a humanistic and libertarian strategy for mental health, emphasizing individual accountability, free will, and self-governance. He believed that people should have the freedom to live their lives as they wish, without being identified as mentally ill or being forced into psychiatric treatment against their will [9,10].

Communication is an essential part of the community, and textual communication is presently one of the most common ways to express ourselves. People utilize social networks to explain their sentiments, mental states, goals, and desires, as well as to document their activities or routines [11]. According to the latest survey reports, more than half of the population (59%) uses social media, and time spent on social media accounts for two-thirds of total internet usage [12]. People often share health information online to gain experience-based knowledge, find emotional support, and work together towards their health objectives. This includes receiving information on specific therapies or behaviors and collaborating on related decisions. Individuals with mental illnesses are more prone to expressing themselves online, whether through blogging, social networking, or public forums [13]. As people write more digitally, an enormous amount of data can be analyzed automatically to infer meaningful information about one's well-being, such as mental health conditions. Using social data provides an additional advantage in reducing the stigma associated with mental health screening, as such techniques can create new chances for early diagnosis and intervention, as well as fresh insights into the study of the causes and processes of mental health [12,13].

Partially due to the stigma surrounding mental illness and the frequent difficulty in receiving appropriate care through the healthcare system, people are increasingly turning to social media to express their problems and find emotional support [14,15]. As the number of people suffering from mental illness is growing, coupled with the pressure imposed on health and social care systems, social media platforms have evolved into a source of "in-the-moment" daily conversation, with subjects such as mental health and well-being being discussed. This presents an interesting research opportunity to better understand and classify various types of mental health through analyses of social media data (Twitter and Reddit) [16].

Traditionally, monitoring mental health was carried out by asking carefully crafted questions to a random sample of the population in order to conduct mental health surveys. However, high-quality survey data require a significant investment of effort, time, and money for survey designers, interviewers who gather data, and participants who voluntarily provide answers [17]. Relying on self-reported statistics (survey data) is also problematic since mental illness is prone to bias.

Social media data provide the following benefits over surveys: (i) Social media data are very inexpensive to gather in comparison to the price of conventional sample surveying, especially when comparing the prices of telephone surveys (phone calls plus interviewee expenses), in-person surveys (interviewee expenses plus probable travel fees), and internet interviews (postage and printing) [18,19]. (ii) Traditional surveys only have a limited ability to observe the respondent's actual behaviors and can only ask the participant about their behaviors; the relationship between responses and real behaviors is often weak. On the other hand, social media offers a plethora of data regarding user behavior because social media postings are made outside of the context of the survey; in other words, social media data offer a record of a user's actual behavior [18,19]. (iii) Social media data provide potentially large samples compared to survey data (on Twitter, more than 500 million tweets are posted every day, and around 1.12 million posts are posted on Reddit) [18], making it possible to access queries about more specific "subgroups" (for example, issues affecting a certain geographical area) [19].

Furthermore, although previous research has demonstrated the feasibility of examining and predicting mental health from social media activities, many such studies are platform-specific. Single-platform analysis may restrict findings and actions to a subset of the target population [20]. Although useful information on a certain behavior or themed community may be available on one platform, such knowledge may not be available on another [21–23]. In addition, there is evidence that shows that individuals behave differently on various social media platforms due to the platform's distinctive social etiquette and design features of the platform [24].

The general objective of a study states what is expected to be achieved in general terms. Specific objectives are:

1. To examine the linguistic characteristics and patterns of different social media activities associated with different mental health groups;
2. To investigate whether a machine learning model can be developed to categorize a user's social media activity patterns into different mental illness groups;
3. To understand if a machine learning model trained on a specific social medium can generalize to other social media platforms.

## 2. Related Work

An increasing number of individuals are using social media platforms such as Twitter, Facebook, Reddit, and Instagram to express themselves and communicate with others in real-time. As a result, vast amounts of social data are created, containing important information about people's interests, emotions, and behaviors [7]. Social media is transforming how people self-identify as having a mental health condition and how they interact with others who have had similar experiences, often asking about treatment, and side effects, and reducing feelings of stigma and loneliness. The study of prominent social media platforms such as Reddit and Twitter may provide insight into what patients are most concerned about (more so than their physicians) [24]. Furthermore, this form of large-scale user-generated content (social media) provides a unique opportunity to study mechanisms underlying mental health disorders. For example, research on children and adolescents has indicated that frequent daily use of social networking platforms is independently associated with poor self-rating of mental health, higher levels of psychological distress, and suicidal thoughts [25,26].

Beyond simple features such as frequency of usage, researchers have now employed more sophisticated methods to extract in-depth usage pattern features such as linguistic style, affective content of the posts, and the interaction pattern as characterized via a social graph to predict mental health conditions associated with specific social media posts [26]. The language used in Reddit forums dedicated to mental health has been studied to discover linguistic traits that might be useful in creating future applications to detect individuals who require immediate assistance [17]. Users' self-disclosure in Reddit mental illness forums has been studied to create language models that explain social support, which has been found to contain informational, emotional, instrumental, and prescriptive information. Even though Redditors are not paid for their work, the feedback expressed in the comments is of remarkably high quality; it can be both emotional and useful, as well as informative. This is a crucial difference compared to social media platforms such as Twitter, where sharing health information is frequently broadcast or an emotional outburst and not always about seeking accurate or detailed information about diagnosis and treatment [26,27].

Social media data have been identified as a resource for gaining knowledge about mental illnesses. For example, Twitter data have been used to develop classifiers that can identify individuals who are depressed [27]. Coppersmith et al. (2015) used Twitter data to identify linguistic characteristics that may be used to classify Twitter users into those suffering from mental illness and those who do not [25]. Dinu et al. (2021) used Reddit data to classify various mental illness groups based on users' posts rather than individual users or groups of users. Supervised machine learning, which is used for categorization or prediction modeling, offers the advantage of accounting for complicated interactions

between variables that were previously unknown [28]. As datasets become larger and variables become more complex, machine learning techniques may become a useful tool in psychiatry to correctly detangle variables linked with patient outcomes [29,30].

Goffman et al. (2009) categorized three types of stigmas present in our society: physical, moral, and tribal. The first type, physical stigma, is based on visible or physical differences, including skin color, disability, and disfigurement. These types of stigmas are easily identified and often lead to social exclusion and discrimination. The second type, moral stigma, is rooted in perceived moral failures or character flaws, such as addiction, criminal behavior, or mental illness. These prejudices are associated with negative stereotypes and often cause social rejection and marginalization. Lastly, tribal stigma is derived from being a member of a specific group or community, such as racial or ethnic minorities, religious groups, or non-heterosexual orientations. Tribal stigmas are a result of social norms and cultural values and can lead to discrimination and exclusion from mainstream society [31].

Stigma is a social construct that denotes a negative perception or attitude towards individuals or groups who deviate from social norms due to factors such as race, gender, sexual orientation, and physical or mental health [32]. Throughout history, stigma has been associated with various social issues. For instance, in ancient times, people with physical disabilities or deformities were often believed to be cursed or possessed by evil spirits. Similarly, individuals with mental health issues were thought to be demon-possessed during the Middle Ages and were subjected to cruel treatments such as exorcism [33]. Initially, researchers viewed stigma as an individual problem, where stigmatized individuals were seen as having a personal deficiency. However, later research adopted a social model, recognizing that stigma is often a result of broader societal attitudes and structural inequalities. This perspective emphasizes the importance of addressing social and structural barriers to reduce stigma [32].

Several researchers have used machine learning on social media data and healthcare for classifying various mental illness G groups. For example, Gkotsis et al. (2016) [16] collected data from 11 different mental health subreddits and developed a multiclass classification model. If a user suffers from several mental health issues, such as anxiety and depression, the user can submit posts in multiple subreddits. If the model is trained on posts from users with multiple symptoms, the multiclass classification model may suffer from being noisy [16]. Kim et al. (2020) collected data across six mental and health-related subreddits and developed six binary classification models for each mental illness (anxiety, autism, bipolar disorder, BPD, depression, and schizophrenia) and utilized pre-trained word vectors rather than random initialization, which yields superior results; however, the classification model suffers from noisy data if the model is trained with the posts of users with multiple symptoms [34].

Numerous studies have shown that language usage, social expressiveness, and interaction are important indicators of mental health. The Linguistic Inquiry Word Count (LIWC), a validated technique for the psychometric evaluation of language data [35], has been used extensively to analyze linguistic features associated with various mental illnesses. For example, De Choudhury, M. et al. (2015) [26] gathered Twitter posts from individuals who had been diagnosed with depression and used the Linguistic Inquiry and Word Count (LIWC) to examine the linguistic and emotional characteristics of the tweets [26]. Coppersmith et al. (2015) [25] also emphasized that associated language patterns, such as the use of first-person pronouns, negative emotions, and angry words, had a significant relationship with mental problems. Several studies examine the relationship between language usage and mental health [25]. According to Aaron Beck et al.'s (1967) cognitive theory of depression, depressed people tend to view themselves and their surroundings negatively. They frequently use negative terms and first-person pronouns while expressing themselves (I, or me) [25]. Rude et al. (2004) [36] examined linguistic patterns of essays written by college students who were depressed, had been depressed in the past, and had never been depressed. His findings showed that depressed students used fewer positive emotion words and more negative valence words [36].

Despite growing interest in the detection of mental illness, present efforts have mostly been limited to research on a single platform, with less emphasis given to generalizability across multiple social media platforms. The reasons why it is important to check if the model can be generalized across different platforms can be summarized as follows:

(1) First, even the most widely used social media platforms are not used by everyone, and most platforms only reach small segments of the population. As an illustration, 25% of US adults claim to use Twitter [37], while 18% of US adults claim to use Reddit. Social media users do not confine themselves to a single social media platform; instead, users efficiently navigate across multiple platforms to express themselves by exploiting variations among these platforms [38].

Furthermore, social media data are skewed in terms of demographics. Twitter, for example, has a 55% overall adoption rate in the United States. However, approximately 38.5% of those aged 25 to 34 use Twitter, with the great majority using it multiple times per day. Furthermore, 57% of respondents claimed their primary motivation for accessing Twitter is to increase their understanding of current events. Reddit has a 39% overall adoption rate in the United States. However, roughly 64% of those aged 18 to 29 use Reddit, with the great majority using it multiple times per day. A total of 72% of respondents claimed their primary motivation for accessing Reddit is for entertainment [39]. There are socio-demographic biases associated with social media and these must be thoroughly investigated before drawing broad generalizations about the broader population. For example, although Reddit has a large user base with a wide range of socio-demographics, with an estimated 6% of internet users active on Reddit, there is a gender bias (8% of male internet users compared to 4% of female). With an estimated 18.7% of internet users active on Twitter, there is a bias towards male users (12.3% of male internet users compared to 6.4% of females) and a bias toward younger users, with a higher percentage of users aged 18–49 than those over 50, on these platforms [40].

(2) Different social media platforms may feature different usage patterns. For instance, Twitter may provide more frequent updates on an event, whereas Reddit may provide more critical analysis regarding the same events. Furthermore, Twitter may discuss political news and current events more rigorously than Reddit, but Reddit may be a better choice for news updates and entertainment discussions. Social media is mostly driven by normal users; therefore, a platform's suitability depends on how the corresponding users utilize it. For example, if a considerable number of people discuss an event, then the event is important. In an emergency, receiving frequent updates is critical; therefore, a platform with active users is better suited for this type of event. Thus, diverse characteristics of the content published, user posting behavior, and post-spreading patterns across these platforms can prove to be useful for meeting certain requirements such as exploring important events, live updates, or analyzing news stories [41].

Therefore, in this study, we investigate two popular social media platforms, Reddit, and Twitter. Specifically, we are interested in examining how different/similar linguistic characteristics and patterns of activities are associated with different mental health groups in the two platforms. Following this, we study how machine learning models trained on one platform are generalizable to another.

## 3. Methodology

This study will be used to develop a model for categorizing various mental illness using social media activity pattern. Figure 1 illustrates phases of a typical classification identification framework: (1) Data collection/extraction, (2) Data pre-processing, and (3) Classification model.

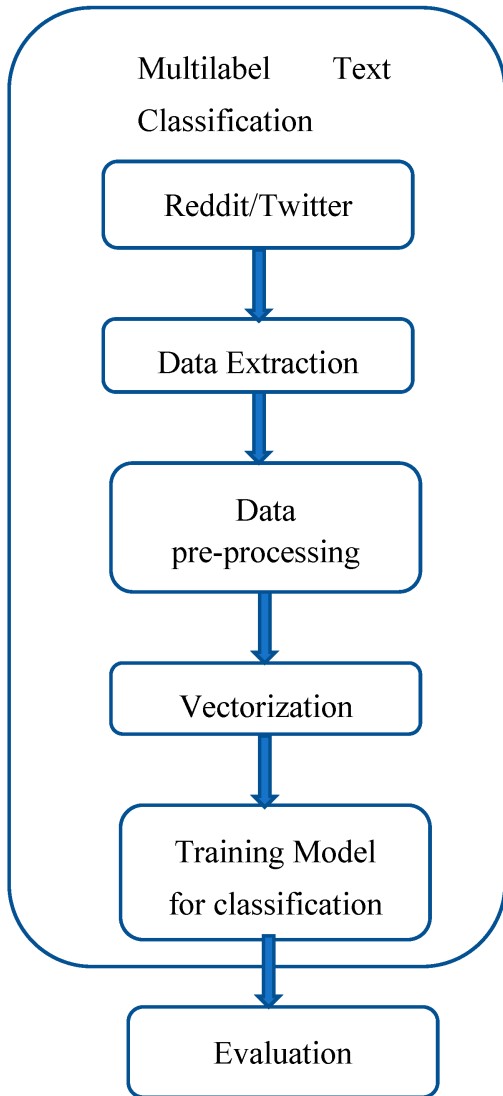

**Figure 1.** Proposed methodology [42].

### 3.1. Data Collection

We used Twitter's Streaming API (Application Programming Interface) to continuously collect Tweets featuring #Depression, #Anxiety, #Bipolar, #BPD (borderline personality disorder), #Schizophrenia, and #Autism between January 2017 and December 2018. The data on the categories "subreddit r/Depression, r/Anxiety, r/Bipolar, r/BPD (borderline personality disorder), r/Schizophrenia, and r/Autism" on the Reddit dataset were obtained from the author [34]. Note that none of the user data contain any personally identifying information because these have all been anonymized. In all, 248,537 people contributed 606,208 posts across the six subreddits and 23,102,773 English tweets across 6 hashtags were extracted from Reddit and Twitter, respectively [34].

### 3.2. LIWC (Linguistic Inquiry and Word Count)

"*LIWC is a transparent text analysis program that counts words in psychologically meaningful categories*" [37]. Pennebaker et al. (2015) describe approximately 90 variables that were analyzed with LIWC. Table 1 provides a list of LIWC2022 dictionary language dimensions from the chosen set of LIWC categories [35].

**Table 1.** Linguistic Inquiry and Word Count Language dimensions and output variable information [39].

| LIWC Variables | Description | Examples | Words/Entries in Category |
|---|---|---|---|
| **Clout** | Relative social position, confidence, or leadership [41] | – | – |
| **Analytical thinking** | Logical, formal, and hierarchical thinking processes [41,42] | – | – |
| **Authenticity** | Perceived honesty, and genuineness [35] | – | – |
| **Emotional tone** | Degree of positive/negative tone [35] | – | – |
| **Pronouns** | Self-presentation and attention, ego, other people, and things, and involvement [43,44]. | I, them, itself | 74/286 |
| **Verbs, Adverbs, and Adjectives** | Content of communication [44] | a, an, the, very, really, and, but, whereas | 1560, 159/514, 1507 |
| **Death** | Suicidal thoughts and related conversation [44,45] | death, dead, die, kill | 109 |
| **Culture** | Politics (political, legal), Ethnicity (racial, ethnic), Technology (scientific and technological) [44] | car, United States, govern, phone | 772 |
| **Lifestyle** | Leisure, work, religion, life, and money [44,46] | work, home, school, work | 1437 |

*3.3. Data Preprocessing*

The dataset input at this stage has various unwanted data that must be removed. Pre-processing steps involve removing punctuation, Twitter-specific terms, special symbols, and numbers to obtain the cleaned data for analysis [47]. This stage usually deals with noisy data. It is necessary to transform some data to make it suitable for analysis, which can be carried out with normalization and attribute derivation methods [47]. The internet is a huge source of data, and in this huge data, there is always some information that is not important and not required for the analysis purpose. Filtering this data is time-consuming and can be achieved using attribute selection and numerous reduction techniques, such as stop words' removal, stemming, and tokenization [47].

Data pre-processing was carried out in our study before model training and evaluation. For each post/Tweet, we deleted unwanted punctuation and utilized space and then applied Python's natural language toolkit (NLTK) to tokenize posts and filter commonly used terms (stop transformed words). Following that, all the words were transformed into lowercase, and stop words were removed. Porter Stemmer, a tool used to define a set of criteria for researching word meaning and source, was used on the tokenized words to convert a word to its root meaning and reduce the amount of word corpus [34]. Since we believe that posts with fewer than 25 characters may not include enough information to be classified, such small posts/Tweets were removed from the mental group to reduce the quantity of the data. Following this process, information from 488,472 posts (Reddit) created by a total of 228,060 people and from 15,932,364 posts from Twitter was used for the analysis [34,48].

*3.4. Classification Models*

The purpose of a classification algorithm is to select appropriate categories from data based on model parameters learned from training data. The classifier then predicts categories for new data [49]. The classification step of the process determines the real mapping between the message and whether it belongs to a specific class (e.g., depression, anxiety, BPD, etc.) or not [50].

Six binary classification models were developed, each of which classifies the posts into one of the following six keywords: depression, anxiety, bipolar disorder, BPD, schizophre-

nia, and autism. The aim was to detect a potential mental health condition by constructing six different models for each mental disease, each of which incorporated data from users who had posted messages about specific mental problems. For constructing a model for identifying depression, for example, we labeled tweets/posts with the depression hashtag/subreddit as a depression class; the rest of the posts were classified as a non-depression class. The data were divided into the following categories: (i) training (80%) and testing (20%) (Reddit); (ii) training (80%) and testing (20%) (Twitter); (iii) training (100% of Reddit data), and testing (100% of Twitter data); and (iv) training (100% of Twitter data) and testing (100% of Reddit data). For the CNN classifier, we used the word2vec API of the Python Package, Genism, to incorporate words from pre-processed texts [29].

Traditional machine learning algorithms primarily employ a bag of words or n-gram techniques to build feature vectors to train classifiers. Since there is a very limited number of words in short texts such as tweets, the traditional machine learning algorithms suffer from issues related to dimensionality and data sparsity. Today, neural networks combined with word embeddings are used for text classification, which has demonstrated a remarkable performance gain [51].

The architecture of a CNN model constitutes an input layer, a 1-Dimension pooling layer, a 1-Dimension convolutional layer, and an output layer. The first layer of the model is an embedding layer that represents the word embeddings of a 20-dimensional pre-processed post, and its weight is set by the pre-trained word2vec. Second, a convolutional layer with word vector input consists of 128 filters. The next layer is a 128-layer max-pooling layer that takes the highest values from the CNN filters. The output of the max-pooling layer is routed via two densely linked layers, with the ultimate output being the probability of classification using the sigmoid activation function, which runs from 0 to 1. The batch size was set to 128, and the training epochs were set to 5 [34].

The dataset was uploaded to Google Colab as a CSV file using Google Drive. The model was trained for about 220 h for classification models that were trained on Twitter and whose performance was tested on Reddit, 180 h for classification models which were trained on Reddit and whose performance was tested on Twitter, 240 h for classification models which were trained and tested on Twitter and 60 h for classification models which were trained and tested on Reddit.

## 4. Results

### 4.1. LIWC Statistical Results

Using the LIWC software, we retrieved linguistic characteristics from the posts/tweets. The LIWC software was used to count the number of corresponding words and categorize them into 90 distinct feature variables using an existing list of words and categories (e.g., personal pronouns, and positive/negative phrases) [52].

Table 2 displays the mean, standard deviation, and *t*-test score of LIWC indicator ratings for the terms on Twitter and Reddit. The average word count for Twitter is 11.9, and for Reddit, it is 198.85. The analytic thinking indicator scores indicated that those on Twitter used more formal and logical terms, whereas those on Reddit used more informal and narrative expressions. The two LIWC indicators' ratings, clout, and authenticity, indicated that the posts on Twitter conveyed their unfavorable sentiments less confidently and personally than people on Reddit. According to the emotional tone indicator ratings, posts on Reddit communicated with more negative expressions than posts on Twitter. However, both scores were lower than 50, indicating that both Twitter and Reddit groups largely communicated using negative sentiments, which is expected given the nature of the discussion topics. The four LIWC indicators' ratings, pronoun, verb, adjective, and conjunctive, indicate that interaction patterns are extremely similar across various subreddits, and hashtags (Reddit and Twitter) are focused on content rather than people [31,53].

**Table 2.** Mean and standard deviation of LIWC indicator across various mental illness disorders.

| | Twitter | | Reddit | | |
|---|---|---|---|---|---|
| | **Mean** | **Std** | **Mean** | **Std** | ***t*-Value** |
| Word count | 11.90 | 6.35 | 198.86 | 234.92 | −2734.1 |
| Analytic | 68.40 | 29.04 | 18.12 | 19.27 | 1199.22 |
| Authentic | 44.93 | 41.04 | 84.79 | 24.02 | −674.01 |
| Big Words | 31.29 | 18.70 | 14.90 | 5.46 | 611.66 |
| Dictionary | 71.99 | 21.34 | 94.84 | 4.29 | −747.82 |
| Tone | 41.33 | 40.58 | 20.7 | 26.54 | 352.19 |
| Death | 0.51 | 2.51 | 0.36 | 1.04 | 23.90 |
| Pronoun | 3.08 | 6.08 | 20.07 | 4.95 | 1927 |
| Verb | 16.67 | 13.85 | 21.46 | 4.81 | −241.17 |
| Adverb | 3.95 | 6.68 | 7.93 | 3.45 | −414.31 |
| Adjective | 9.93 | 9.88 | 6.28 | 3.07 | 257.72 |
| Male Reference | 0.68 | 2.91 | 0.69 | 1.68 | −0.86 |
| Female Reference | 0.55 | 2.68 | 0.69 | 1.72 | −35.96 |
| Negative emotion | 2.69 | 5.94 | 2.68 | 2.62 | 1.02 |
| Conjunction | 0.70 | 2.65 | 7.61 | 2.85 | 810.33 |
| Clout | 47.6 | 30.98 | 11.30 | 22.27 | 291.2 |

A *t*-value, also referred to as a t-statistic, is a statistical measure that assesses the distinction between the means of two samples, taking into account the variability in the data. Formula (1), used to calculate the *t*-value, is as follows:

$$t = (X1 - X2)/(s * \sqrt{(1/n1 + 1/n2)}) \tag{1}$$

where X1 and X2 are the means of two independent samples being compared and s is the pooled standard deviation (2).

$$s = \sqrt{\frac{\left((n1-1) * s1^2 + (n2-1) * s2^2\right)}{(n1 + n2 - 2)}} \tag{2}$$

where s1 and s2 are the standard deviations of the two samples, and n1 and n2 are the sample sizes of the two samples.

To compute it, the difference between the means of the two samples is divided by the standard error of the difference. Applying a two-sample *t*-test shows that the differences between Reddit and Twitter are statistically significant (*p*-value < 0.005). The only exceptions, as observed, are the "male" categories and "negative emotions".

The posts (Twitter and Reddit) consisted of encouragement words (positive sentiment) such as "friend", "love", "work", "Today", "great", "time", "today", "great", "life", "think", "right" and "life", whereas negative sentiment words were related to "stress", "mental", and "anxiety" (Table 3). Our findings revealed that "time" was a popular topic of conversation, and it was related to "family", "friend", and "love". Individuals may receive comfort and support from social media relationships with family and friends while keeping hopeful that they will soon spend time together. The words "home" and "work" connected with the word "life" were frequently brought up by those who were working from home or unemployed. These results show that most discussions revolved around work-related problems. There was additional talk about "people", "worry", and "stress", suggesting discussions were focused on how to overcome mental illness [54]. The word cloud in Figure 2 represents the frequency of words that occur in the text feature from Reddit users' post descriptions related to mental illness. The word cloud in Figure 3 represents the frequency of words that occur in the text feature from Twitter users' post descriptions related to mental illness.

**Table 3.** The most frequently used word stems from Twitter and Reddit users' post descriptions related to mental illness.

| | Twitter | | | | Reddit | | |
|---|---|---|---|---|---|---|---|
| **Word** | **Frequency** | **Rows with Word** | **% of Rows with Word** | **Word** | **Frequency** | **Rows with Word** | **% of Rows with Word** |
| Amp | 940,080 | 710,242 | 9.27 | Time | 371,710 | 191,927 | 41.10 |
| people | 883,407 | 772,094 | 10.08 | Life | 317,105 | 163,829 | 35.08 |
| Time | 621,330 | 573,703 | 7.49 | People | 285,856 | 149,326 | 31.98 |
| Day | 556,455 | 492,958 | 6.44 | Year | 278,438 | 148,516 | 31.80 |
| Life | 498,948 | 461,547 | 6.03 | Day | 267,515 | 149,157 | 31.94 |
| Love | 425,265 | 382,664 | 5.00 | Friend | 254,676 | 123,898 | 26.53 |
| Give | 421,597 | 402,717 | 5.26 | Work | 228,378 | 122,824 | 26.30 |
| Work | 410,108 | 376,156 | 4.91 | Anxiety | 204,225 | 105,729 | 22.64 |
| Year | 396,209 | 360,660 | 4.71 | Start | 199,621 | 118,414 | 25.36 |
| See | 393,048 | 369,749 | 4.83 | Talk | 182,367 | 105,836 | 22.66 |
| Talk | 333,709 | 306,045 | 4.00 | See | 172,339 | 111,272 | 23.83 |
| Great | 327,726 | 304,222 | 3.97 | Thought | 168,401 | 106,154 | 22.73 |
| Today | 317,043 | 301,611 | 3.94 | Love | 133,905 | 76,625 | 16.41 |
| mental | 294,956 | 275,864 | 3.60 | Month | 123,387 | 84,456 | 18.09 |
| school | 288,989 | 261,007 | 3.41 | Job | 119,284 | 62,546 | 13.39 |
| stress | 284,533 | 259,233 | 3.38 | Week | 119,033 | 80,745 | 17.29 |

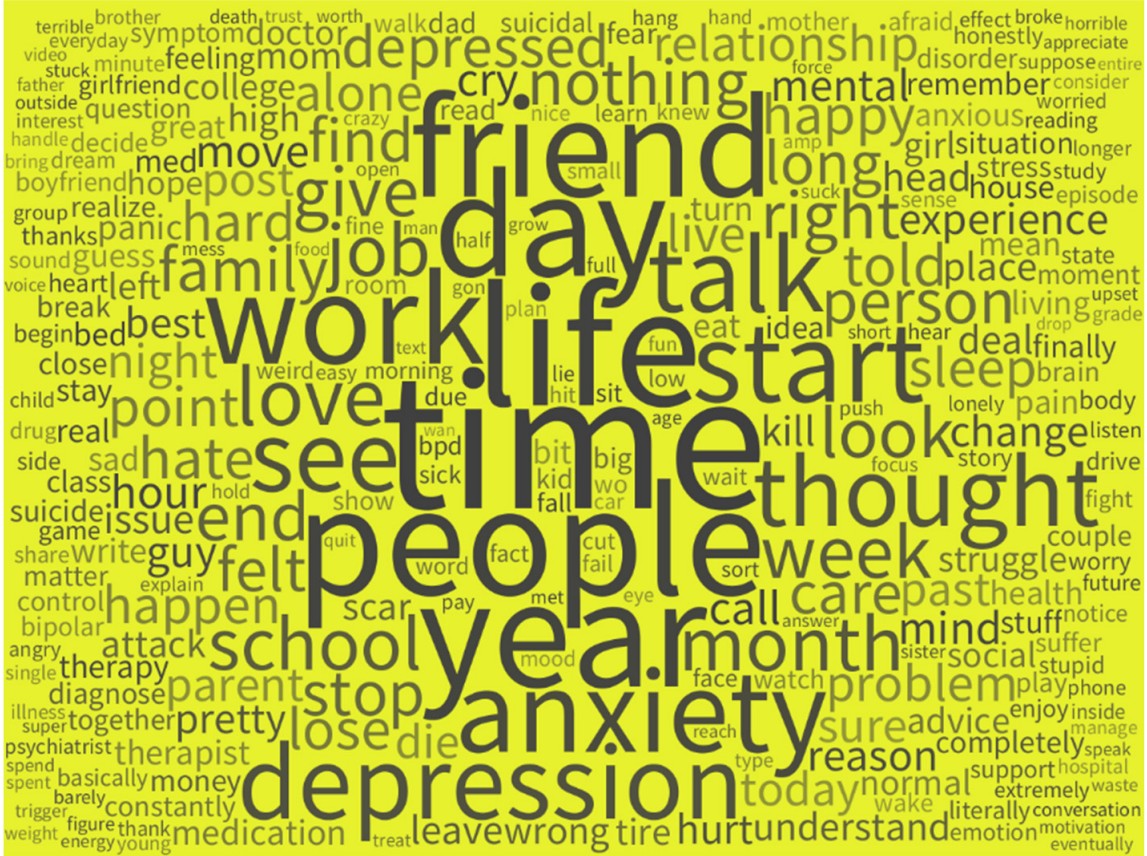

**Figure 2.** Word cloud of most frequently used word stems from Reddit users' post descriptions related to mental illness.

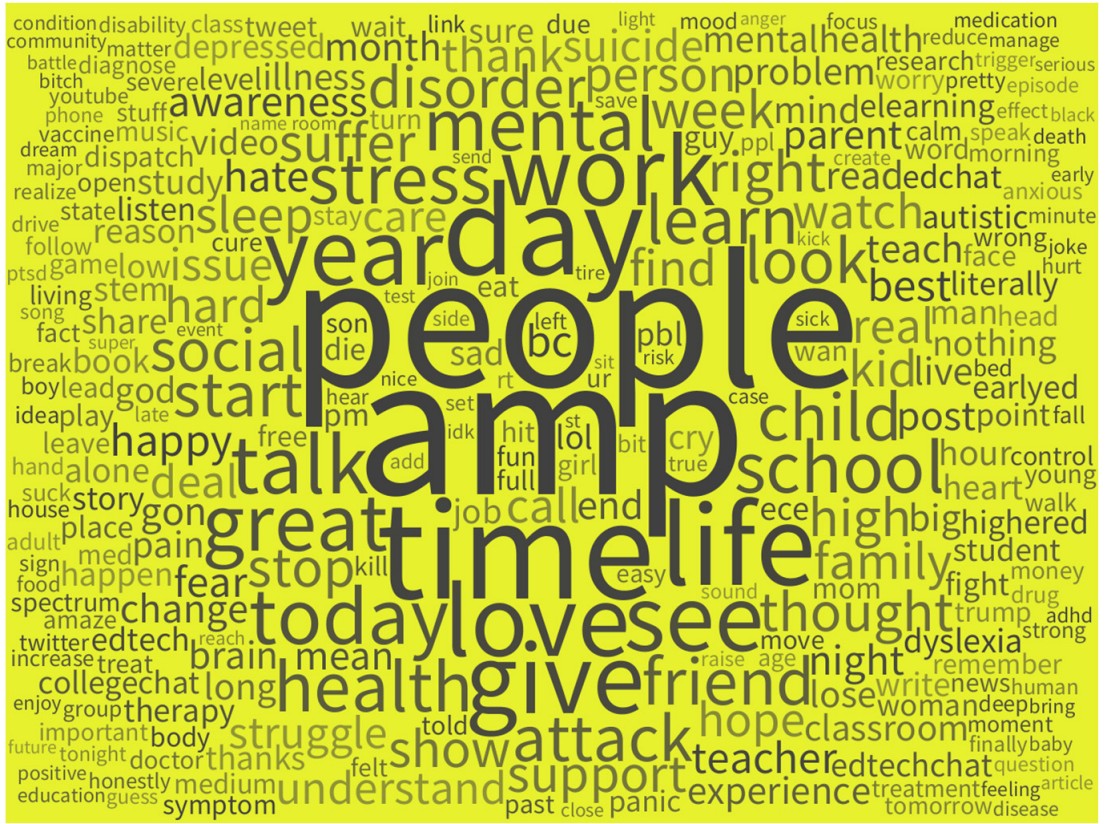

**Figure 3.** Word cloud of most frequently used word stems from Twitter users' post descriptions related to mental illness.

*4.2. Machine Learning Model Evaluation*

We employed the typical metrics for evaluating machine learning models: *accuracy (3)*, *precision (4)*, *recall (5)*, and *F1-Score (6)*. The following are the definitions of these metrics:

$$Accuracy = TN + TN/(TP + FP + FN + TN) \tag{3}$$

$$Precision = TP/(TP + FP) \tag{4}$$

$$Recall = TP/(TP + FN) \tag{5}$$

$$F1 - score = 2 * (Recall * precsion)/(Recall + Precision) \tag{6}$$

where *TP* is truly positive, *FP* is a false positive, *TN* is a true negative and *FN* is a false negative.

Table 4 highlights the performance of six binary classification models which were trained on Twitter and tested on Twitter. Among the six posts, autism had the best accuracy (96.67%) on CNN. Anxiety, schizophrenia, depression, and BPD also demonstrated high accuracy with CNN models, 84.53%, 85.85%, 84.53%, and 92.14%, respectively, and their F1-scores in identifying mental illnesses were greater than 80%.

**Table 4.** Model evaluation of convolutional neural network of six binary classification models which were trained and tested on Twitter.

| Label | | Precision | Recall | F1-Score | Accuracy | Source |
|---|---|---|---|---|---|---|
| Autism | 0 | 97.23 | 98.34 | 97.78 | 96.67 | |
| | 1 | 90.37 | 83.58 | 86.84 | | |
| Anxiety | 0 | 86.23 | 87.48 | 86.85 | 84.53 | |
| | 1 | 84.71 | 84.83 | 84.77 | | |
| BPD | 0 | 93.42 | 92.91 | 93.16 | 92.14 | |
| | 1 | 92.47 | 80.22 | 85.91 | | |
| Bipolar | 0 | 97.32 | 98.54 | 97.93 | 95.89 | Twitter |
| | 1 | 96.48 | 96.21 | 96.34 | | |
| Schizophrenia | 0 | 85.42 | 86.83 | 86.12 | 85.85 | |
| | 1 | 57.46 | 55.47 | 56.45 | | |
| Depression | 0 | 83.87 | 86.53 | 85.18 | 84.32 | |
| | 1 | 84.42 | 88.68 | 86.50 | | |

Table 5 highlights the performance of six binary classification models which were trained on Reddit and performance-tested on Reddit. Among the six different posts, schizophrenia had the highest accuracy (96.6%) on CNN. The other posts, anxiety, autism, bipolar disorder, depression, and BPD, also demonstrated high accuracy with CNN models, 91.89%, 91.33%, 95.34%, 87.21%, and 81.12%, respectively, and their F1 scores in identifying mental illnesses ranged above 50%.

**Table 5.** Model evaluation of convolutional neural network of six binary classification models which were trained and tested on Reddit.

| Label | | Precision | Recall | F1-Score | Accuracy | Source |
|---|---|---|---|---|---|---|
| Autism | 0 | 94.73 | 95.34 | 95.03 | 91.33 | |
| | 1 | 59.34 | 55.68 | 57.45 | | |
| Anxiety | 0 | 95.52 | 95.69 | 95.60 | 91.89 | |
| | 1 | 77.33 | 75.58 | 76.44 | | |
| BPD | 0 | 86.23 | 87.94 | 87.08 | 81.12 | |
| | 1 | 83.41 | 84.67 | 84.04 | | |
| Bipolar | 0 | 96.32 | 98.23 | 97.27 | 95.34 | Reddit |
| | 1 | 71.91 | 61.19 | 66.12 | | |
| Schizophrenia | 0 | 98.72 | 96.43 | 97.56 | 95.88 | |
| | 1 | 64.28 | 49.64 | 56.02 | | |
| Depression | 0 | 85.23 | 77.93 | 81.42 | 87.21 | |
| | 1 | 83.32 | 83.51 | 83.41 | | |

Table 6 highlights the performance of six binary classification models which were trained on Twitter and performance-tested on Reddit. Among the six different posts, schizophrenia had the best accuracy (94.49%) on CNN. Autism, bipolar disorder, and BPD also demonstrated high accuracy with CNN models, 88.42%, 83.79%, and 86.93%, respectively, and their F1 scores in identifying mental illnesses ranged from the twenties to sixties percent, which were lower than those with class-balanced channels. Table 7 highlights the performance of six binary classification models which were trained on Reddit and tested on Twitter. Among the six models, autism had the best accuracy (97.42%) on CNN. Schizophrenia and BPD also demonstrated high accuracy with CNN models (96.35% and 89.71%, respectively), and their F1 scores in identifying mental illnesses ranged from the thirties to the sixties percent, which were lower than those with class-balanced channels.

**Table 6.** Model evaluation of convolutional neural network of six binary classification models which were trained on Twitter and whose performance was tested on Reddit.

| Label | | Precision | Recall | F1-Score | Accuracy | Source |
|---|---|---|---|---|---|---|
| Autism | 0 | 87.53 | 98.44 | 92.66 | 88.42 | |
| | 1 | 94.62 | 19.84 | 32.80 | | |
| Anxiety | 0 | 56.43 | 85.42 | 67.96 | 59.24 | |
| | 1 | 70.33 | 34.41 | 46.21 | | |
| BPD | 0 | 95.32 | 95.79 | 95.55 | 86.93 | Train—Twitter |
| | 1 | 32.28 | 26.89 | 29.34 | | Test—Reddit |
| Bipolar | 0 | 97.31 | 86.23 | 91.44 | 83.79 | |
| | 1 | 28.83 | 27.58 | 28.19 | | |
| Schizophrenia | 0 | 90.31 | 96.5 | 93.30 | 94.49 | |
| | 1 | 81.6 | 24.39 | 37.55 | | |
| Depression | 0 | 56.82 | 65.93 | 61.04 | 56.12 | |
| | 1 | 58.33 | 48.61 | 53.03 | | |

**Table 7.** Model evaluation of convolutional neural network of 6 binary classification models which were trained on Reddit and whose performance was tested on Twitter.

| Label | | Precision | Recall | F1-Score | Accuracy | Source |
|---|---|---|---|---|---|---|
| Autism | 0 | 98.43 | 98.28 | 98.35 | 97.42 | |
| | 1 | 35.78 | 53.23 | 42.79 | | |
| Anxiety | 0 | 84.41 | 81.79 | 83.08 | 72.29 | |
| | 1 | 44.34 | 46.72 | 45.50 | | |
| BPD | 0 | 90.21 | 98.6 | 94.22 | 89.71 | Train—Reddit |
| | 1 | 91.34 | 31.63 | 46.99 | | Test— |
| Bipolar | 0 | 92.61 | 77.47 | 84.37 | 73.12 | Twitter |
| | 1 | 58.23 | 48.91 | 53.16 | | |
| Schizophrenia | 0 | 97.42 | 94.84 | 96.11 | 96.35 | |
| | 1 | 69.14 | 32.31 | 44.04 | | |
| Depression | 0 | 51.31 | 61.57 | 55.97 | 52.23 | |
| | 1 | 52.78 | 42.52 | 47.10 | | |

As shown in Table 8, the classic CNN-based Word2Vec [34] text classification approach had an average precision, recall rate, F1-score, and accuracy of 82.7%, 69.4%, 71.83%, and 89.4%, respectively. The classification algorithms in the current study had an average precision, recall rate, F1-score, and accuracy of 82.5%, 79.5%, 80.5%, and 91.5%, respectively.

**Table 8.** Comparison of model evaluation of convolutional neural network which was trained and tested on Reddit.

| Class | XGBoost [29] | | CNN [29] | | | | Proposed Methodology | | | |
|---|---|---|---|---|---|---|---|---|---|---|
| | F1-Score | Accuracy | Precision | Recall | F1-Score | Accuracy | Precision | Recall | F1-Score | Accuracy |
| **0** | 78.65 | 71.69 | 89.1 | 71.75 | 79.49 | 75.13 | 94.73 | 95.34 | 95.03 | 91.33 |
| **1** | 58.02 | | 58.66 | 82.04 | 68.41 | | 59.34 | 55.68 | 57.45 | |
| **0** | 77.73 | 70.41 | 87.54 | 41.44 | 56.25 | 77.81 | 95.52 | 95.69 | 95.6 | 91.89 |
| **1** | 55.92 | | 75.92 | 96.91 | 85.14 | | 77.33 | 75.58 | 76.44 | |
| **0** | 91.93 | 85.53 | 87.22 | 38.02 | 52.95 | 90.2 | 86.23 | 87.94 | 87.08 | 81.12 |
| **1** | 53.39 | | 90.4 | 99.05 | 94.53 | | 83.41 | 84.67 | 84.04 | |
| **0** | 91.37 | 85.14 | 91.84 | 32.69 | 48.21 | 90.49 | 96.32 | 98.23 | 97.27 | 95.34 |
| **1** | 46.43 | | 90.42 | 99.54 | 94.76 | | 71.91 | 61.19 | 66.12 | |
| **0** | 92.52 | 86.72 | 81.16 | 24.87 | 38.07 | 94.33 | 98.72 | 96.43 | 97.56 | 95.88 |
| **1** | 40.97 | | 94.62 | 99.56 | 97.03 | | 64.28 | 49.64 | 56.02 | |
| **0** | 97.35 | 94.91 | 48.08 | 49.39 | 48.73 | 96.96 | 85.23 | 77.93 | 81.42 | 87.21 |
| **1** | 38.31 | | 98.48 | 98.4 | 98.44 | | 83.31 | 83.51 | 83.41 | |

Figure 4 highlights the accuracy of six binary classification models: Twitter model, Reddit model, Twitter model tested on Reddit, Reddit model tested on Twitter.

### Classification accuracy of various mental illness

**Figure 4.** Classification accuracy of various mental illness.

1.  To examine the linguistic characteristics and patterns of different social media activities associated with different mental health groups.

    Findings 1: The discussions (encouragement words (positive sentiment)) on both social media platforms (Twitter and Reddit) focus on how people obtain support from family and friends and discuss problems regarding mental illness, which are a common topic of discussion among individuals suffering from mental illness [54].
    Findings 2: Both Twitter and Reddit groups largely communicated using negative sentiments, which is expected given the nature of the discussion topics [54–56].

2.  To investigate whether a machine learning model can be developed to categorize a user's social media activity patterns into different mental illness groups.

    We applied a state-of-the-art traditional machine learning model (CNN + word2vec) for successfully classifying various mental illnesses.
    Findings 1: In terms of mental health classification, the Reddit model compared to the Twitter model performed better for anxiety, schizophrenia, and depression and has the lowest F1-score on autism, bipolar disorder, and schizophrenia.
    Finding 2: Kim, J., et al.'s (2020) [34] model achieved a remarkable accuracy. Our suggested model is simpler and has a lower level of complexity analysis; however, it achieves a greater level of accuracy (2.11%). A critical step was taken to optimize the model by tuning the hyperparameter to understand if a machine learning model trained on a specific social media can generalize to other social media platforms.

3.  We applied a state-of-the-art traditional machine learning model (CNN and word2vec) to provide a generalized method for successfully categorizing various mental illnesses using social media data (Twitter and Reddit).

    Finding 1: The Reddit model tested on Twitter compared to the Reddit model tested on Reddit performed better for autism, BPD, and schizophrenia. The Twitter model tested on Twitter compared to the Twitter model tested on Reddit performed better for anxiety, autism, schizophrenia, bipolar disorder, and depression.
    Finding 2: In terms of mental health condition classification, the Reddit model tested on Twitter compared to the Twitter model tested on Reddit performed better for autism, anxiety, BPD, and schizophrenia. We compared the results of testing the Twitter model on Reddit and testing the Reddit model on Twitter to better understand how well each model

generalizes beyond the platform. The Reddit model seems to have a better performance than the Twitter model, indicating that the Reddit model can generalize better to another (Twitter) social media platform.

## 5. Discussion

1.  To examine the linguistic characteristics and patterns of different social media activities associated with different mental health groups.

The mental illness lexicon on social media platforms such as Twitter and Reddit was analyzed using LIWC. The similarities are as follows: (i) both Twitter and Reddit groups largely communicated using negative sentiments, which is expected given the nature of the discussion topics [54–56]; (ii) the discussions (encouragement words) (positive sentiment) on both social media platforms (Twitter and Reddit) focused on how people obtain support from family and friends and discussed problems regarding mental illness, which is a common topic of discussion among individuals suffering from mental illness [54]. However, there are some differences. Computing the mean and standard deviation across LIWC indicators shows that the differences between Reddit and Twitter are statistically significant when comparing a chosen set of LIWC categories (clout, analytical thinking, authenticity, tone, pronoun, death, emotion). This could be due to the restriction on words, or the depth of topics discussed on Twitter as compared to Reddit. According to the literature, length influences writing style, exhibiting specific linguistic aspects; for example, length limits disproportionately retain negative emotions, adverbs, and articles, and conjunctions have the highest probability of being omitted [57].

2.  To investigate whether a machine learning model can be developed to categorize a user's social media activity patterns into different mental illness groups.

In this study, the machine learning model (CNN + word2vec) has been successfully used to classify various mental illnesses using social media platforms (Twitter and Reddit). We are comparing the proposed model, which was trained and tested on Reddit, with Kim et al.'s (2020) model [34]. Our proposed approach has a greater classification effect, as well as a higher average recall rate, F1 score, and accuracy rate. Kim, J., et al. (2020) reported that the model achieved remarkable accuracy. However, our suggested model is simpler and has a lower level of complexity analysis, achieving a greater level of accuracy (2.11%). A critical step was taken to optimize the model by tuning the hyperparameter. The Reddit model has the lowest F1-score on autism, bipolar disorder, and schizophrenia, which is due to the class imbalance problem [34].

The reasons why the performance of Reddit is better compared to that of Twitter are as follows: (i) Reddit communities are monitored by individuals who volunteer to be moderators. Moderating privileges include the ability to delete posts and comments from the community. A moderated Reddit post might become a safe space to discuss topics related to mental health. Reddit users may find it more comforting that a moderator may remove harsh or harmful messages or individuals from the subreddit [58,59]. (ii) Because there are more posts with promotional content on Twitter, tweets about mental illness symptoms are sometimes diluted by other topics such as fitness blogs or meditation seminars. [58,59].

3.  To understand if a machine learning model trained on a specific social media platform can be generalized to other social media platforms.

We investigated machine learning classifiers (CNN and word2vec) to provide a generalized method for classifying various mental illnesses using social media data (Twitter and Reddit). We trained and tested machine learning models using labeled Twitter datasets, and then compared the performance of our trained models to other social media sources using non-Twitter datasets (Reddit). Despite the differences in linguistic characteristics between Reddit and Twitter, our machine learning models could generalize the model between social media platforms (Twitter and Reddit). We compared the results of testing the Twitter model on Reddit and testing the Reddit model on Twitter to better understand how well

each model generalizes beyond the platform. The Reddit model seemed to have a better performance than the Twitter model, indicating that the Reddit model could generalize better compared to another social media platform (Twitter). The Reddit model tested on Twitter had F1 scores ranging from the twenties to sixties percent across various mental health conditions and had the lowest accuracy for depression and anxiety. The Twitter model tested on Reddit had F1 scores ranging from thirties to sixties percent across various mental illnesses and had the lowest accuracy for depression, which is due to the class imbalance problem [34].

The reasons why Reddit is better compared to Twitter are as follows: (i) We believe that the reason the Reddit model is better at generalization is due to the interaction structure of Reddit making it ideal for seeking expert opinions. Twitter may provide more frequent updates on an event, whereas Reddit may provide more critical analysis regarding the same events. Furthermore, Twitter users tend to discuss political news and current events more so than Reddit users; however, Reddit may be a better choice for news updates and entertainment discussions [41]. Twitter is suitable for obtaining frequent updates during an emergency or any live event. Reddit's unrestricted post length plays a vital function in providing us with additional background information. The reasons why Twitter may be better compared to Reddit are as follows: (ii) In contrast to Reddit, imposing a post length constraint on Twitter helps to reduce biased and extreme viewpoints. Furthermore, an event on Twitter can be tracked for a longer period, which might be valuable for analyzing its evolution [41]. (iii) Twitter's high negative associativity suggests that information dispersion is heavily influenced by the user who created the tweet. When a user has many followers, their post is more likely to spread quickly and widely. Reddit users do not have close communities, as seen by a low clustering coefficient, and a small number of related components; as a result, information spreads slowly on Reddit on average [41]. Thus, there are significant differences between these two platforms (Twitter and Reddit) in terms of user behavior as well as their conversation and posting patterns. However, given the available abundance of these platforms, each with its uniqueness in presentation, spreading patterns, and user interests, a comparative analysis of their efficacy would be beneficial [41].

Our long-term objective is not only to provide aid to clinical researchers and policy-makers in addressing communications on social media but also to aid individuals suffering from mental illness. To allocate resources more effectively and provide help where it is most needed, the authorities (policymakers, healthcare professionals, etc.) must keep track of the population's mental health over time and across different geographic regions. Our findings might be beneficial for policymakers, academics, and healthcare professionals interested in understanding the occurrence of different mental health conditions and concerns over time and in different locations, and hence in formulating better policies, recommendations, and health promotion activities in response to address the issue [16].

## 6. Conclusions and Future Scope

We present a method for automatically identifying social media (Twitter and Reddit) posts related to mental health and then classifying them into theme-based groups (subreddit and hashtag) using a machine-learning algorithm. Our research shows that users on the two different platforms have different themes of interest and different sentiments, indicating the need to examine cross-platform social media networks to offer a more comprehensive view of people's opinions. Rather than relying on a single platform, the integration of various online social networks (OSNs) can assist stakeholders (politicians and healthcare professionals) in gaining a more thorough understanding of community responses [60].

This study aims to categorize a user's social media data (Twitter and Reddit) into different mental illness groups. In this study, we analyzed machine learning's generalization capacity to classify various mental illnesses across multiple social media platforms (Twitter and Reddit). The effectiveness of our machine learning models (CNN and word2vec) was demonstrated even when they were trained on texts without the keywords "depression",

"anxiety", "bipolar", "bpd", etc. It is significant to note that even when tested on datasets unrelated to the training datasets, the machine learning approach (CNN and word2vec) proposed in this study performed well.

We plan to extend our analysis to other types of social media content (e.g., multi-media content) related to mental illness. Furthermore, we would be able to undertake a comprehensive investigation of the impact of multiple datasets on the results obtained. Online social media data can "fill in the gaps" by providing continuous, real-time measurements of a wide spectrum of people's thoughts and emotions. However, as depressed people may stop posting on social media, testing for continuing monitoring apps should also be carried out using other uninterrupted data sources, such as smartphone and sensor data. It is also necessary to conduct studies that combine social media data with clinical interviews and other screening techniques in ecologically valid samples to evaluate the incremental value of social media-based screening and differentiating between mental health conditions [61].

## 7. Limitation

There are a few limitations to this study. While our study offers a thorough grasp of the generalization capacities of social media classifiers for mental health, we acknowledge that there is still more to learn. Only two platforms and six mental health conditions methods are taken into consideration in this study. In the future study, more platforms can be used for analysis. The proposed study does not consider several factors (e.g., age, gender, regional differences, etc.) that could influence classification models. These parameters can be taken into consideration in future studies to enhance the quality or accuracy of deep learning models. This is needed to investigate the effect of regional differences on mental health and observe and check mental health considering various demographics. We could use an ensemble approach with our various binary classification models to identify co-morbid illnesses and other real-world mental issues.

**Author Contributions:** Conceptualization, C.S.A. and R.V.; methodology, C.S.A. and R.V.; software, C.S.A. and R.V.; validation, C.S.A. and R.V.; formal analysis, C.S.A. and R.V.; investigation, C.S.A. and R.V.; resources, C.S.A. and R.V.; data curation, C.S.A. and R.V.; writing—original draft preparation, C.S.A. and R.V.; writing—review and editing, C.S.A.; visualization, C.S.A.; supervision, C.S.A.; project administration, C.S.A. All authors have read and agreed to the published version of the manuscript.

**Funding:** This research received no external funding.

**Institutional Review Board Statement:** Not applicable.

**Informed Consent Statement:** Not applicable.

**Data Availability Statement:** Data will be shared based upon request through the corresponding author.

**Conflicts of Interest:** The authors declare no conflict of interest.

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
