# Peer review of "Generalizability of Machine Learning to Categorize Various Mental Illness Using Social Media Activity Patterns"

_societies, doi:10.3390/soc13050117_

Round 1

Reviewer 1 Report

Thanks to the editor for giving me the opportunity to review this article.

The aim of this paper is to find out if machine learning can accurately classify various types of mental illness in social media by analyzing users' posts—especially when those posts 9 don't explicitly contain certain terms (keywords) like "depression" or "anxiety," etc.

General comment

 The article in general presents a sufficient structure. The theme of social media has now become excessively fashionable, in fact there is a certain similarity with other papers in the literature. The article suffers from some limitations such as that of not having analyzed gender issues regarding the perception of mental illness between men and women and how gender and the process of acquiring gender roles influence the ways in which the disease is manifested on social media. That would have made it much more interesting. 

Furthermore, talking about mental illness without citing the classics of sociological literature (Goffman, for example) makes the article scarce from a theoretical point of view. Exactly the same also when it comes to language and not even one author belonging to symbolic interactionism has been mentioned. Another major limitation of the study concerns the lack of reference to the biopsychosocial paradigm of the disease. The article is also very scarce in terms of references, they need to be implemented in a meaningful way.

Specific comments

Abstract: is confusing and overly long. It is recommended to make it more fluent and concise.

Introduction: The authors should first introduce the concept of health and illness according to a biopsychosocial approach and then move on to the definition of mental illness. Furthermore, reference should also be made to the sociological approach which does not accept the psychiatric classification of mental illnesses (see Thomas Szasz). When the authors mention stigma it is necessary to mention Goffman and his classification of the different types of stigma present in society with a focus on how the concept of stigma has changed over time.

Methodology: Why didn't the authors resort to netnography? Furthermore, the study carried out, from a methodological point of view, would be a discrete study (an approach suggested by Webb, Lee, etc.). I suggest assuming to change the type of methodology. It would also be advisable to deepen the selection criteria of the words used in the search.

Results: The results suffer from the lack of analysis of gender issues, and are presented in a too aseptic way, I suggest adding graphs, if possible, some sociograms to make the results less difficult for the reader to follow.

Discussion: Since this is an article with a much-studied approach in the literature, it is necessary to greatly implement the discussion with greater comparisons with other studies, possible similarities in the use of some variables or not.

It is necessary to introduce a paragraph "Strengths and limitations" to explain in detail all the limitations that this study has (reporting many of the ones I have suggested above).

Conclusions and Future scope: these sections have a sufficient structure, although in the light of the comments they will need to be expanded and commented on in greater depth.

English needs to be extensively revised.

Reviewer 2 Report

Paper titled (Using machine learning approaches to categorize various mental illness groups using social media activity patterns) discussed the utility of a machine based approach for a social purpose; it was used to categorize mental illness groups in social media. The aim is straightforward and clear.

1- Title should be more informative and give impression about the results

2- Introduction is ling and need to be shortened to be more concrete

3- Figure 1 in methodology was drawn in not elegant and informative way, 

4- Table 1: title should be clear and gives information with no  abbreviation

5- Method in general acks refernces & especially for equations

6- Nothing is Bipolar: should be Bipolar disoreder, through out the manuscript.

7- Did the authors classify the age of the population affected with these diseases?

8- Authors did not write on how statistical analysis was done? T value how was calculated?

9-Use appropriate abbreviations for hours, seconds, and other units etc

10- Tables: use the same degree of precision in numbers, 2 decimal places is enough or three & fix it in all results

11- please Improve the appearance of the tables according to MDPI guide for authors

Round 2

Reviewer 1 Report

Now the paper is very good. Best regards

Reviewer 2 Report

The revised version of paper titled (Using machine learning approaches to categorize various mental illness groups using social media activity patterns) was revised partly but some important points were not corrected as recommended & authors did not highlight where in the text & page, we can find these corrections. Give a point to point answers with mention for the place in the text for the following questions:

1- Authors did not write on how statistical analysis was done? T value how was calculated?

2- Tables: use the same degree of precision in numbers, 2 decimal places is enough or three & fix it in all results. 

Titles should be above the table itself. not contienous with the previous text.

Round 3

Reviewer 2 Report

the revised version of paper titled (Using machine learning approaches to categorize various mental illness groups using social media activity patterns) is now acceptable for publication